# Carbon dioxide capture and functionalization by bis(*N*-heterocyclic carbene)-borylene complexes

Jun Fan[1,4], An-Ping Koh[1,4], Chi-Shiun Wu[2], Ming-Der Su[2,3] & Cheuk-Wai So [1]✉

Derivatives of free monocoordinated borylenes have attracted considerable interest due to their ability to exhibit transition-metal-like reactivity, in particular small molecules capture. However, such complexes are rare as the formation is either endergonic, or the resulting adduct is a transient intermediate that is prone to reaction. Here, we present the synthesis of two bis(*N*-heterocyclic carbene)-borylene complexes capable of capturing and functionalizing carbon dioxide. The capture and subsequent functionalization of $CO_2$ by the bis(NHC)-disilylamidoborylene **1** is demonstrated by the formation of the bis(NHC)-isocyanatoborylene-carbon dioxide complex **3**. Reversible capture of $CO_2$ is observed using the bis(NHC)-mesitylborylene **2**, and the persistent bis(NHC)-mesitylborylene-carbon dioxide adduct **4** can be stabilized by hydrogen bonding with boric acid. The reactions of **4** with ammonia-borane and aniline demonstrate that the captured $CO_2$ can be further functionalized.

The conversion of carbon dioxide ($CO_2$) into value-added chemicals has attracted much attention due to the increasing amount of anthropogenic $CO_2$ in the atmosphere and consequent climatic problems[1]. Due to the high thermodynamic stability of $CO_2$, reactive precious transition metal complexes have been developed to capture, activate, and transform $CO_2$ into high-value chemical feedstocks, but some of these elements remain costly and susceptible to potential supply chain issues[2–5]. In this context, the development of sustainable alternatives that possess energetically accessible molecular orbitals to interact with $CO_2$ is important.

Boron derivatives with both strong electrophilic and nucleophilic characters were selected to examine their feasibility in transition-metal-like small-molecules activation. Braunschweig et al. showed that multiply bonded diboron compounds such as bis(NHC)-diborene (NHC = *N*-heterocyclic carbene) and bis(CAAC)-diboracumulene (CAAC = cyclic (alkyl)(amino)carbene) could activate $CO_2$ via cycloaddition of B=B and partial B≡B bond with $CO_2$, respectively (Fig. 1a)[6]. We further illustrated that the B=B double bond in an *N*-phosphinoamidinato NHC-diborene complex was capable of catalyzing hydroboration of $CO_2$ with HBpin[7]. Kinjo et al. reported that the

detached nucleophilic and electrophilic boron centres in a 6π-aromatic 1,3,2,5-diazadiborinine functioned like frustrated Lewis pairs (FLPs) to cooperatively activate $CO_2$[8], which enabled the latter to undergo catalytic *N*-formylation with amines and HBpin (Fig. 1a)[9]. Wilson and Gilliard et al. showed that the CAAC ligand and boron anion in a 9-CAAC-9-borafluorene anion cooperatively activated two equivalents of $CO_2$ to form a trioxaborinanone as a perceivable carbon monoxide releasing molecule (Fig. 1a)[10]. Wang and Mo et al. reported that the silylene ligand and borylene center in a distorted T-shaped bis(silylene)amidoborylene cooperatively activated $CO_2$ and 9-BBN in hydroboration via a presumed 2-sila-4-boraoxetan-3-one intermediate (Fig. 1a)[11].

In transition metal-mediated $CO_2$ activation, the first step involves simple coordination of $CO_2$ with a transition metal in the $\eta^2$-$CO_2$, $\eta^1$-$CO_2$-$\kappa C$ or $\eta^1$-$CO_2$-$\kappa O$ binding mode (Fig. 1b)[12]. Several stable transition metal complexes of $CO_2$ such as Aresta's $[(Cy_3P)_2Ni(\eta^2$-$CO_2)]$[13], Herskowitz's $[(diars)_2M(\eta^1$-$CO_2$-$\kappa C)Cl]$ (M = Ir, Rh; diars = *o*-phenylenebis(dimethylarsine))[14] and Gambarotta's $[(ONNO)V(OH)(\eta^1$-$CO_2$-$\kappa O)]$ (ONNO= $[2,4$-Me$_2$−2-(OH)C$_6$H$_2$CH$_2]_2$N(CH$_2)_2$NMe$_2)$[15] were isolated and structurally characterized. However, in boron

[1]School of Chemistry, Chemical Engineering and Biotechnology, Nanyang Technological University, Singapore 637371, Singapore. [2]Department of Applied Chemistry, National Chiayi University, Chiayi 60004, Taiwan. [3]Department of Medicinal and Applied Chemistry, Kaohsiung Medical University, Kaohsiung 80708, Taiwan. [4]These authors contributed equally: Jun Fan, An-Ping Koh. ✉e-mail: CWSo@ntu.edu.sg

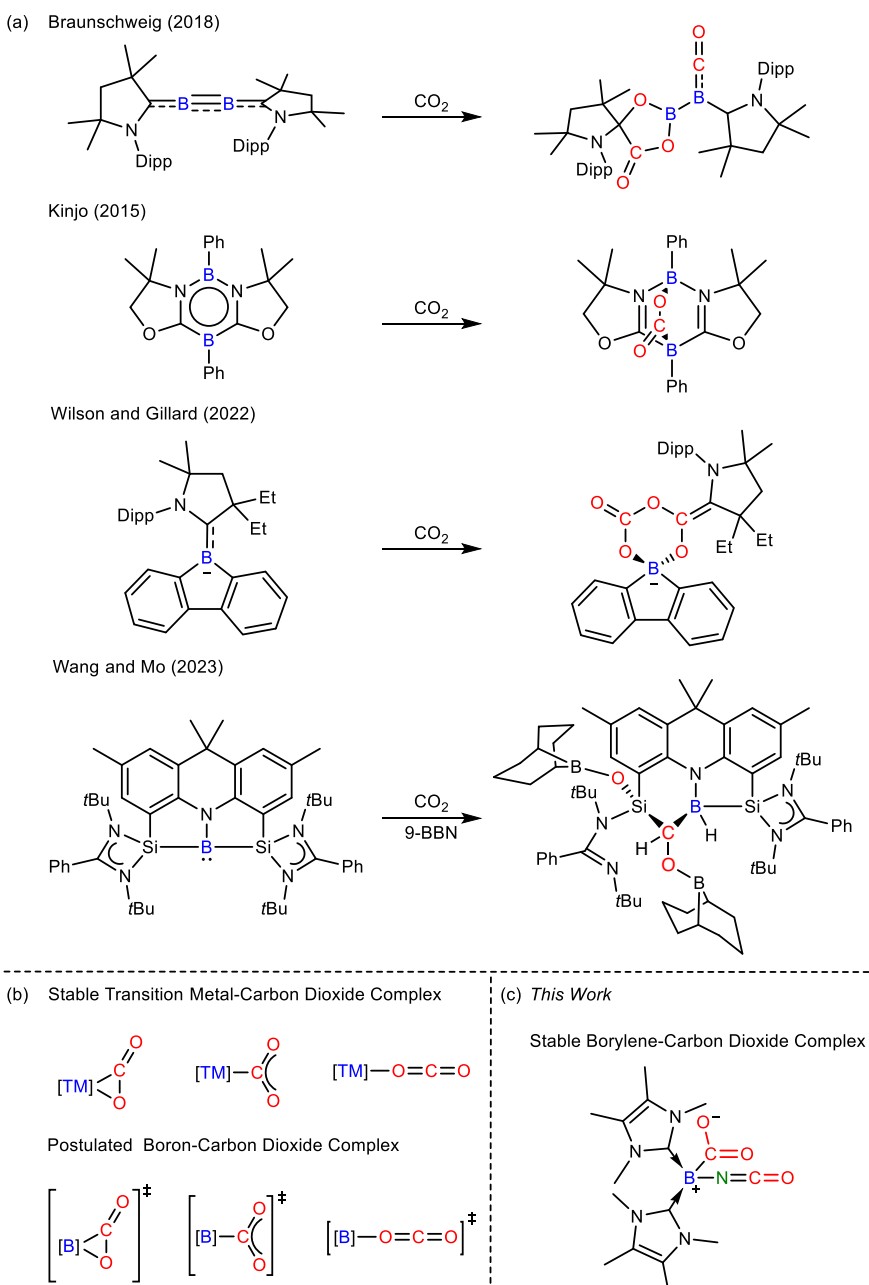

**Fig. 1 | Ambiphilic boron-mediated carbon dioxide activation and its related complexes. a** Carbon dioxide activation by various ambiphilic boron compounds reported previously. **b** Established structures of stable transition metal-carbon dioxide complexes and postulated analogues based on boron. **c** The first stable borylene-carbon dioxide adduct, bis(NHC)-isocyanatoborylene-carbon dioxide adduct **3**.

metallomimetic chemistry, single-site-boron complexes of $CO_2$ with the composition of $B(\eta^2\text{-}CO_2)$, $B(\eta^1\text{-}CO_2\text{-}\kappa C)$ or $B(\eta^1\text{-}CO_2\text{-}\kappa O)$ are unknown. In the activation of $CO_2$ by the 9-CAAC-9-bora-fluorene anion, bis(silylene)amidoborylene or bis(CAAC)-diboracumulene[6,10,11], Density Functional Theory (DFT) calculations show that the nucleophilic boron center captures $CO_2$ in the $\eta^1\text{-}CO_2\text{-}\kappa C$ or $\eta^2\text{-}CO_2$ binding mode (Fig. 1b). However, the reaction is either endergonic, or the resulting adduct is a transient intermediate that is prone to react with the electrophilic ligand backbone or the electrophilic boron center. In $CO_2$ activation mediated by FLP of phosphine and borane[16], the phosphine acts as a Lewis base to capture $CO_2$ in the $\eta^1\text{-}CO_2\text{-}\kappa C$ binding mode while the electrophilic boron center stabilizes the captured $CO_2$ via O-coordination[17]. In this context, a stable single-site-boron-carbon dioxide adduct is a highly sought-after compound

not only for scientific curiosity, but also for a better understanding of how a single-boron center captures $CO_2$ and enables the latter to further react with substrates for functionalization. In this paper, we report the synthesis of a bis(1,3,4,5-tetramethylimidazol-2-ylidene)-bis(-trimethylsilyl)amidoborylene and -mesitylborylene and their complexes of $CO_2$. The functionalization of the captured $CO_2$ is also reported. DFT calculations were performed to elucidate electronic structures.

## Results

Two equivalents of 1,3,4,5-tetramethylimidazol-2-ylidene (IMe) were reacted with $RBBr_2$ [R = N(SiMe_3)_2 and mesityl (2,4,6-Me_3C_6H_2)] and $KC_8$ in toluene at room temperature to afford the bis(NHC)-dis-ilylamidoborylene [(IMe)_2B{N(SiMe_3)_2}] (**1**, Yield: 73%) and bis(NHC)-

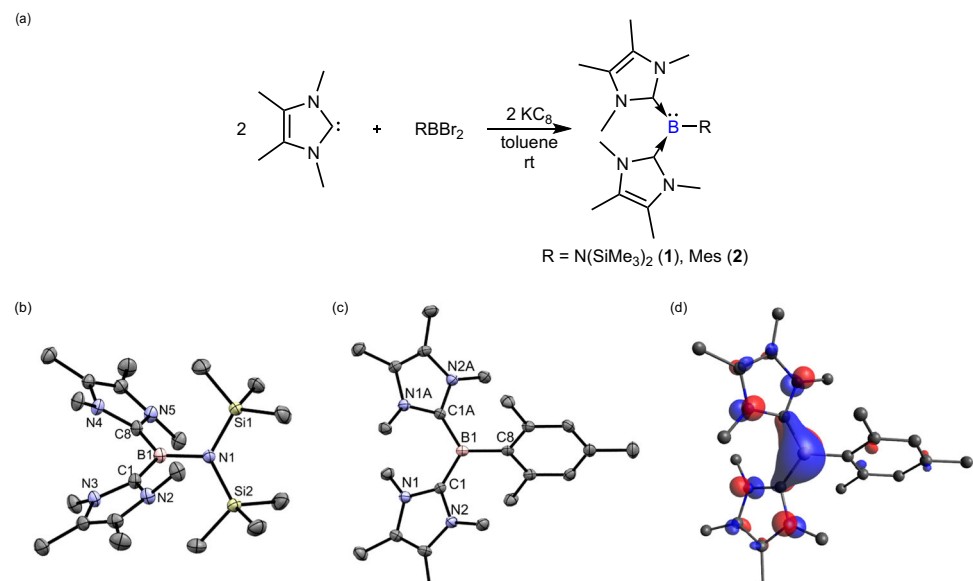

**Fig. 2 | Synthetic routes to compounds 1-2, the molecular structures of 1-2, and the HOMO of 2. a** Synthetic routes to the bis(NHC)-borylenes **1** and **2**. **b** The molecular structure of **1** obtained by X-ray crystallography. Thermal ellipsoids are shown at 50% probability. All hydrogen atoms are removed for clarity. Selected bond lengths (Å) and angles (deg): B1-C1 1.508(2), B1-C8 1.509(2), B1-N1 1.5327(19), C1-B1-C8 119.28(13), C1-B1-N1 120.63(13), C8-B1-N1 120.09(13). **c** The molecular structure of **2** obtained by X-ray crystallography. Thermal ellipsoids are shown at 50% probability. All hydrogen atoms are removed for clarity. Selected bond lengths (Å) and angles (deg): B1-C1 1.520(3), B1-C1A 1.519(3), B1-C8 1.588(5), C1A-B1-C8 120.99(15), C1-B1-C8 120.99(15), C1-B1-C1A 118.0(3). **d** HOMO (−3.865 eV) of **2** (isovalue 0.06) at M06-2X/def2-TZVP level of theory.

mesitylborylene [(IMe)$_2$BMes] (**2**, Yield: 42%, Fig. 2a), respectively. They were both isolated as a red crystalline solid from the concentrated reaction mixture. The $^{11}$B{$^1$H} NMR signals of **1** and **2** are 1.6 and −6.8 ppm, respectively. The molecular structures of compounds **1** and **2** obtained by X-ray crystallography show that the boron centers adopt a trigonal planar geometry (Fig. 2b, c). IMe ligands are tilted with respect to the boron centers. The B-C$_{IMe}$ bond lengths in compound **1** (1.508(2)−1.509(2) Å) and **2** (1.519(3)−1.520(3) Å) are almost equal. They are between the B-C$_{mesityl}$ bond (1.588(5) Å) in **2** and B = C double bonds in methylene boranes (1.351−1.488 Å)[18], indicating some multiple bond character in the B-C$_{IMe}$ bonds due to weak $p_B{\to}p_C$ $\pi$-back bonding. In addition, compound **1** has a gauche conformation with respect to the B1-N1 bond. The Si-N-B-C dihedral angle is 58(1)°. The B-N$_{amido}$ bond (1.5327(19) Å) in **1** is typical of a single bond. DFT calculations (M06-2X/Def2-TZVP) of compounds **1** and **2** show that the Highest Occupied Molecular Orbitals (HOMOs) are a dominant $\pi$-type lone pair orbital on the boron center forming $\pi$-back bonding with the vacant $p$ orbital on IMe, while the Lowest Unoccupied Molecular Orbitals (LUMOs) are the empty $p$ orbitals on the carbene centers (Fig. 2d, Supplementary Figs. 33, 34). Their HOMOs are of similar energy level (**1**: −3.801; **2**: −3.865 eV), indicating that their nucleophilicity should be comparable. The Wiberg Bond Index (WBI) shows that the B-C$_{IMe}$ bonds in compounds **1** (WBI: 1.183−1.185) and **2** (WBI: 1.174−1.176) have weak double bond character with reference to the B-C$_{mesityl}$ bond in compound **2** (WBI: 0.949). The WBI of the B-N$_{amido}$ bond in compound **1** is 0.789, which suggests that the B1-N1 bond has a single bond character. The Natural Population Analysis (NPA) charge of the boron center in compound **1** (0.251 e) is higher than that in compound **2** (−0.055 e) due to the inductive effect exerted by the disilylamido substituent. The trend is in line with the $^{11}$B{$^1$H} NMR chemical shift of compounds **1** and **2**.

Synthesis of stable NHC-borylene complexes is a formidable challenge because the weak $\pi$-accepting property of NHC is insufficient to stabilize the Lewis ambiphilicity of the borylene centers. Robinson and Braunschweig et al. independently showed that NHC-borylene complexes are highly reactive, wherein dimerization or C-H

bond activation often occurred[19,20]. In the case of bis(NHC)-borylene complexes where there are two weak $\pi$-accepting NHC ligands, it is important that the third ligand is strongly $\pi$-electronic withdrawing for the stabilization of the B lone pair of electrons. Braunschweig et al. reported a bis(NHC)-borylene analogue [(I$i$Pr)$_2$BCym] (Cym = (C$_5$H$_4$) Mn(CO)$_3$, I$i$Pr =:C{N($i$Pr)C(H)}$_2$), wherein the lone pair of electrons in the presumed borylene center is stabilized by the B to Mn charge transfer via the Cym ligand leading to a borafulvenium or boratafulvene electronic structure[21]. Driess et al. reported a bis(NHC)-(isocyanide)-borylene cation [(I$i$PrMe)$_2$B(CNR)]$^+$ (I$i$PrMe =:C{N($i$Pr) C(Me)}$_2$; R = cyclohexyl, tert-butyl), where excess electron density on the boron center delocalizes to the isocyanide ligand to afford a prominent boraketiminium resonance form[22]. As NHCs alone are insufficient to stabilize a borylene, strong $\pi$-accepting donors such as cyclic (alkyl)(amino)carbenes (CAACs)[23,24], carbon monoxide or isocyanides[25] were often used to extensively delocalize the boron lone pair of electrons in other stable bis(Lewis base)-borylene complexes. In contrast, compounds **1** and **2** are rare bis(NHC)-borylene complexes that do not need an extra $\pi$-electronic withdrawing substituent to stabilize the borylene centers. In addition, the weak $\pi$-acidity of NHC should preserve the nucleophilic character of borylenes for capturing CO$_2$ and subsequently forming stable single-site-boron complexes of CO$_2$.

Compound **1** was reacted with CO$_2$ (1 bar) in toluene at room temperature, from which the bis(NHC)-isocyanatoborylene-carbon dioxide adduct [(IMe)$_2$(OCN)B($\eta^1$-CO$_2$-$\kappa$C)] (**3**, Fig. 3) was isolated as a colorless crystalline solid (Yield: 80%). The $^{11}$B{$^1$H} NMR signal (−16.0 ppm) supports that the boron center is four-coordinate. The molecular structure of **3** obtained by X-ray crystallography (Fig. 4a) shows that the B-C$_{IMe}$ (B1-C1: 1.640(3); B1-C8: 1.639(3) Å) and B-C$_{CO2}$ (1.645(3) Å) bond lengths are typical of single bonds, while the identical C-O bond lengths of the captured CO$_2$ (1.263(3) and 1.265(3) Å) indicate the presence of delocalized negative charge. The HOMO of **3** in DFT calculations shows the B-C $\sigma$ orbital formed by the lone pair orbital on the B atom and the $\pi^*$ orbital on CO$_2$, leading to a carboxylate anion electronic structure (Fig. 4b). For the formation of **3**, it is proposed that the boron lone pair of electrons attack CO$_2$ and the activated CO$_2$

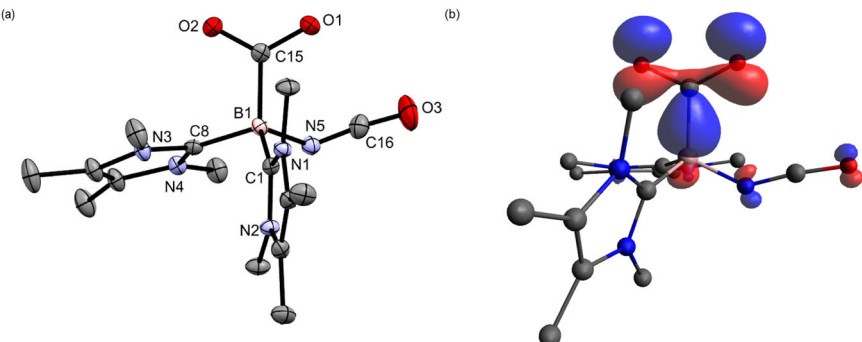

**Fig. 3 | Synthesis of 3 and proposed mechanism for its formation.** Reaction of **1** with $CO_2$ in toluene at room temperature. (DFT calculations, $\Delta G$ and $\Delta G^\ddagger$ in the proposed mechanism are in kcal/mol and were calculated at the M06-2X/def2-TZVP/ IEFPCM(toluene) level of theory).

**Fig. 4 | The molecular structure and HOMO of 3. a** The molecular structure of **3** obtained by X-ray crystallography with thermal ellipsoids shown at 50% probability. All hydrogen atoms are removed for clarity. Selected bond lengths (Å) and angles (deg): B1-C15 1.645(3), C1-B1 1.640(3), C8-B1 1.639(3), B1-N5 1.541(3), C8-B1-C1 106.55(17), C1-B1-N5 105.65(18), N5-B1-C8 111.13(17), O1-C15-O2 123.8(2). **b** HOMO (−6.329 eV) of **3** (isovalue 0.06) at M06-2X/def2-TZVP level of theory, showing the interaction of the lone pair orbital on the B and the $\pi^*$ orbital on $CO_2$.

moiety in **Int1** inserts into the N-Si bond of the disilylamido substituent to form a carbamate substituent in **Int2**. It further captures $CO_2$ to form **Int3**, where the activated $CO_2$ attacks the N-Si bond of the carbamate substituent to form the isocyanate and trimethylsilyl carboxylate substituent in **Int4** and $Me_3SiO^-$. The latter reacts with the trimethylsilyl carboxylate substituent in **Int4** to form compound **3** and $O(SiMe_3)_2$. DFT calculations (M06-2X/def2-TZVP/IEFPCM(toluene)) show that the mechanism is feasible.

Compound **3** is the first stable borylene-carbon dioxide adduct[26]. The formation of **3** demonstrates that the boron center in a borylene can directly attack the carbon center of $CO_2$, which is a result that has remained unattainable using electrophilic borane. The formation of

the isocyanate substituent in compound **3** suggests that functionalization of captured $CO_2$ should be feasible. In this context, compound **2** was used to mediate the functionalization of $CO_2$ due to the presence of a spectated Mes substituent.

The reaction of compound **2** with $CO_2$ (1 bar) in $CD_3CN$ at room temperature afforded the persistent bis(NHC)-mesitylborylene-carbon dioxide adduct [(IMe)$_2$(Mes)B($\eta^1$-$CO_2$-$\kappa$C)] (**4**, $^{11}B\{^1H\}$ NMR: −15.7 ppm, Fig. 5a). When the reaction mixture was placed under reduced pressure or heated at 70 °C, compound **4** was instantaneously converted back into compound **2** as confirmed by $^{11}B\{^1H\}$ NMR spectroscopy, showing that the $CO_2$ capture was reversible. Isolating compound **4** by recrystallization was not attained due to its instability.

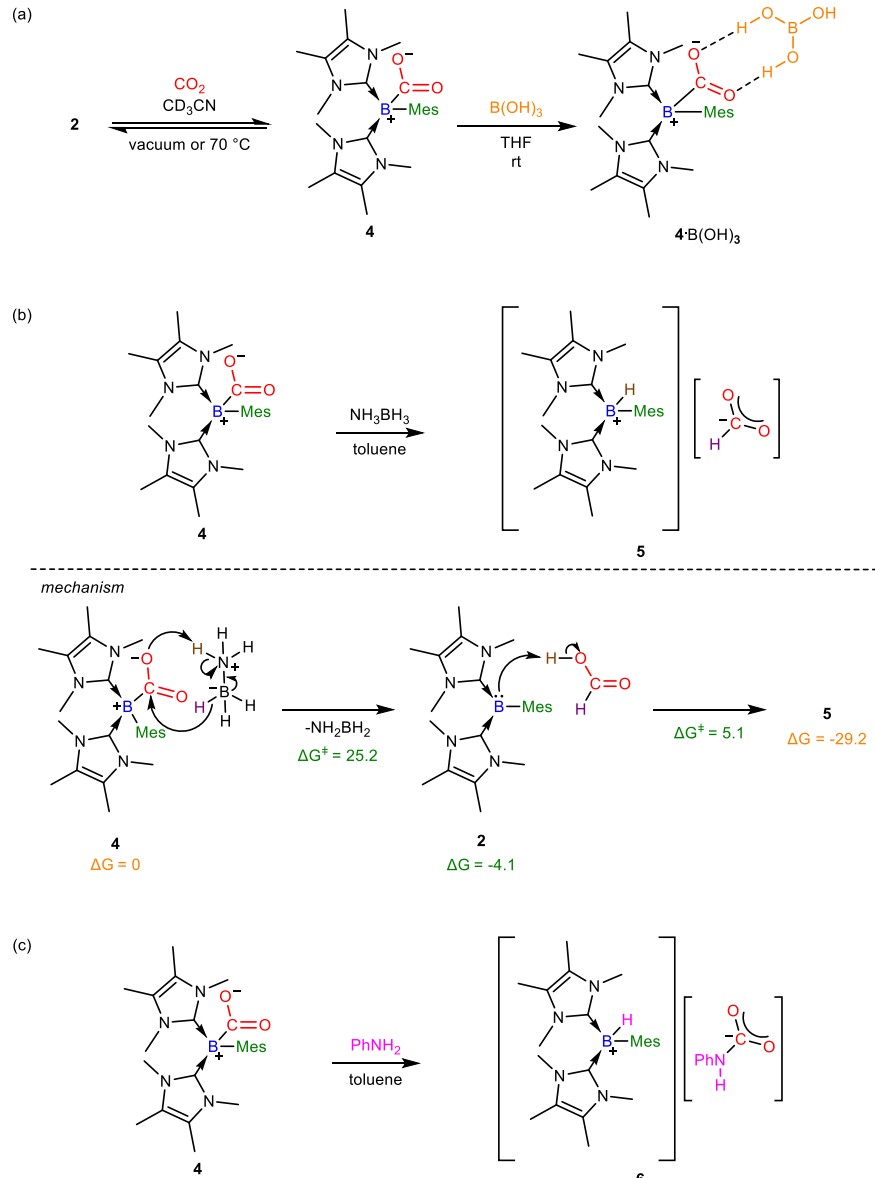

**Fig. 5 | Synthetic routes to compounds 4-6 and the proposed mechanism for the formation of 5. a** Synthesis of **4** and **4·B(OH)$_3$** from **2**. **b** Reaction of **4** with NH$_3$BH$_3$ in toluene at room temperature to form **5**. (DFT calculations, ΔG and ΔG$^‡$ in the proposed mechanism are in kcal/mol and were calculated at the M06-2X/def2-TZVP/ IEFPCM(toluene) level of theory). **c** Reaction of **4** with PhNH$_2$ in toluene at room temperature to afford **6**.

In carbon monoxide dehydrogenase, CO$_2$ is captured by the nucleophilic nickel(0) and electrophilic iron(II) centers, and the activated CO$_2$ substrate is further stabilized by hydrogen-bonding with aptly situated amino acid residues[27]. Based on this, an effective single-site catalyst motif, where ligands with pendant proton donors are used to coordinate with a transition metal for the activation of CO$_2$ through the stabilization of transition metal-CO$_2$ adduct by hydrogen-bonding, has been developed in biomimetic artificial CO$_2$ reduction catalysis[2,28]. It is thus anticipated that the captured CO$_2$ in compound **4** could be stabilized by hydrogen-bonding by mimicking carbon monoxide dehydrogenase. Boric acid B(OH)$_3$ was used to react with compound **4** in THF at room temperature to afford compound **4·B(OH)$_3$** (Fig. 5a), where the captured CO$_2$ moiety was stabilized by hydrogen-bonding with B(OH)$_3$. Compound **4·B(OH)$_3$** was stable in solution under reduced pressure and was isolated as a colorless crystalline solid (Yield: 53%) from the concentrated reaction mixture. The $^{11}$B{$^1$H} NMR spectrum of compound **4·B(OH)$_3$** shows a signal at −15.8 ppm attributable to the mesityl-bonded boron center, which is comparable with that of compound **4**. The

molecular structure of compound **4·B(OH)$_3$** obtained by X-ray crystallography shows that the C-O bonds (1.2643(14), 1.2790(14) Å) are unequal and the oxygen atoms point to two OH substituents of B(OH)$_3$ (Fig. 6a). The O···H distances (O2···H5: 1.77 Å; O1···H3: 1.76 Å) indicate the presence of hydrogen bonding. The B-C$_{IMe}$ (B1-C1: 1.6532(16); B1-C8: 1.6514(17) Å) and B-C$_{CO2}$ bonds (1.6810(17) Å) are typical of single bonds. The B-C$_{mesityl}$ bond (1.6608(17) Å) in compound **4·B(OH)$_3$** is significantly lengthened in comparison with that of compound **2**, probably due to the steric congestion at the four-coordinate boron center.

Borylene-mediated hydrogenation of CO$_2$ with H$_2$ or transfer hydrogenation agents is unknown as yet. As the captured CO$_2$ in compound **4·B(OH)$_3$** is capable of interacting with the hydrogen atoms of B(OH)$_3$, hydrogenation of the captured CO$_2$ should be feasible. As such, NH$_3$BH$_3$ was used to undergo hydrogenation with compound **4** in toluene at 25 °C to form a formate [(IMe)$_2$(Mes)BH](HCO$_2$) (**5**, Yield: 51%, Fig. 5b), which was isolated as a colorless crystalline solid from the concentrated reaction mixture. The $^{11}$B NMR signal of −23.3 ppm (doublet) supports the formation of a B-H bond. The molecular

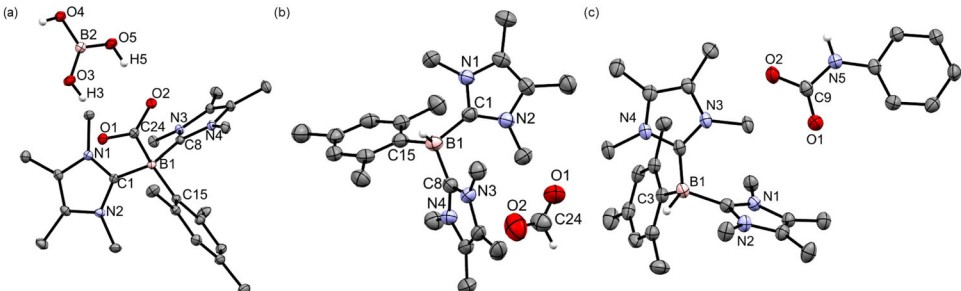

**Fig. 6 | The molecular structures of 4·B(OH)₃, 5 and 6. a** The molecular structure of **4**·B(OH)₃ obtained by X-ray crystallography. Thermal ellipsoids are shown at 50% probability. All hydrogen atoms except for those on boric acid are removed for clarity. Selected bond lengths (Å) and angles (deg): C1-B1 1.6532(16), C8-B1 1.6514(17), C15-B1 1.6608(17), C24-B1 1.6810(17), C24-O1 1.2790(14), C24-O2 1.2643(14), C1-B1-C8 114.24(9), C1-B1-C15 112.85(9), C8-B1-C15 106.30(9), O1-C24-O2 121.60(11), O1-C24-B1 119.45(10), O2-C24-B1 118.68(10). **b** The molecular structure of **5** obtained by X-ray crystallography. Thermal ellipsoids are shown at 50% probability. All hydrogen atoms except for those on the boron and formate are removed

for clarity. Selected bond lengths (Å) and angles (deg): C1-B1 1.635(4), C8-B1 1.621(4), C15-B1 1.622(4), C24-O1 1.223(4), C24-O2 1.210(4), O1-C24-O2 133.6(4), C1-B1-C15 117.2(2), C1-B1-C8 108.4(2), C8-B1-C15 116.3(2). **c** The molecular structure of **6** obtained by X-ray crystallography. Thermal ellipsoids are shown at 50% probability. All hydrogen atoms except for those on the boron and carbamate are removed for clarity. Selected bond lengths (Å) and angles (deg): C1-B1 1.618(3), C4-B1 1.631(3), C3-B1 1.643(3), C9-O1 1.246(2), C9-O2 1.268(2), C9-N5 1.406(2), C1-B1-C4 111.65(15), C1-B1-C3 115.09(16), C3-B1-C4 111.65(15), O1-C9-O2 126.14(18), O1-C9-N5 119.53(17), O2-C9-N5 114.33(17).

structure of compound **5** obtained by X-ray crystallography shows that the boron center adopts a tetrahedral geometry (Fig. 6b), which is consistent with the upfield ¹¹B NMR signal. The OCO skeleton is bent and the C-O bond lengths (1.223(4) and 1.210(4) Å) are shorter than those of **3**, indicating the formation of a formate anion. It is proposed that the hydride from the -BH₃ moiety of NH₃BH₃ attacks the carbon atom of the captured $CO_2$, while the negatively charged oxygen atom of the captured $CO_2$ in compound **4** abstracts a proton from the -NH₃ moiety of NH₃BH₃ to form formic acid HC(O)OH and regenerate compound **2**. The borylene center in compound **2** then activates the O-H bond of HC(O)OH to form compound **5**. The feasibility of this proposed mechanism is demonstrated by DFT calculations (M06-2X/def2-TZVP/IEFPCM(toluene)). Compound **2** is the first borylene capable of mediating hydrogenation of captured $CO_2$ by NH₃BH₃ in Lewis-acid-free conditions to form a formate derivative. With the aid of both sterically hindered Lewis acid and base, Stephan et al. showed that hydrogenation of $CO_2$ with NH₃BH₃ was mediated by an FLP mechanism. In the reaction, the FLP of Mes₃P and AlX₃ (X = Cl, Br) activated $CO_2$ to form [Mes₃PC(OAlX₃)₂], which subsequently reacted with NH₃BH₃ and H₂O to form methanol via a postulated intermediate [Mes₃PH][(MeO)ₙAlX₄₋ₙ][29]. Instead of using NH₃BH₃, Dyson and Corminboeuf et al. reported that the FLP of tris(p-bromo)tridurylborane (tbtb) and 1,8-diazabicyclo[5.4.0]undec-7-ene (DBU) activated H₂ under high pressure to form [DBU·H]⁺[H·tbtb]⁻, which reacted with $CO_2$ via hydride transfer to afford the formate salt [DBU·H]⁺[HCOO]⁻ and regenerate tbtb[30]. The molecular structure of the formate salt [DBU·H]⁺[HCOO]⁻ is similar to that of compound **5**.

Aromatic amines cannot react with $CO_2$ at ambient conditions to form carbamic acids or carbamates[31]. Conversely, the captured $CO_2$ in compound **4** could react with PhNH₂ in toluene at room temperature to form the carbamate [(IMe)₂(Mes)BH][PhN(H)CO₂] (**6**, Yield: 72%, Fig. 5c), which was isolated as a colorless crystalline solid from the concentrated reaction mixture. The ¹¹B NMR signal of compound **6** is −23.4 ppm (doublet) and its X-ray crystallographic data is consistent with compound **5** (Fig. 6c). The formation of compound **6** illustrates another example of further functionalization of the captured $CO_2$ in compound **4**.

## Discussion

This work reports the synthesis of bis(NHC)-disilylamidoborylene **1** and bis(NHC)-mesitylborylene **2** through the reaction of two equivalents of IMe with RBBr₂ (R = N(SiMe₃)₂, 2,4,6-Me₃C₆H₂) and KC₈. Compounds **1** and **2** are examples of rare bis(NHC)-borylene complexes that do not need an extra π-electronic withdrawing substituent to stabilize the borylene centers. The weak π-acidity of NHC preserves

the nucleophilic character of the borylenes and enables the capture of $CO_2$ in the form of the bis(NHC)-isocyanatoborylene-carbon dioxide adduct **3**, which is the first stable single-site-boron complex of $CO_2$. Reversible coordination with $CO_2$ was also demonstrated using compound **2** to form the persistent bis(NHC)-mesitylborylene-carbon dioxide adduct **4**, which was stabilized by hydrogen bonding with B(OH)₃ to form compound **4**·B(OH)₃. Compound **4** was found to be able to undergo hydrogenation with NH₃BH₃ to form formate **5** and amination with PhNH₂ to form carbamate **6**, which demonstrates that the captured $CO_2$ can be further functionalized.

## Methods

### General procedures

All operations were carried out under an inert atmosphere of argon gas by standard Schlenk techniques. The synthesis of the starting materials (TMS)₂NBBr₂ and MesBBr₂ were adapted from published procedures, which can be found below. All other chemicals were purchased from Sigma-Aldrich and used directly without further purification. All solvents were dried over K metal or CaH₂ prior to use. The ¹H, ¹¹B, ¹¹B{¹H}, ¹³C{¹H}, and ²⁹Si{¹H} NMR spectra were recorded on a JEOL ECA 400 spectrometer or Bruker Avance III 400. The NMR spectra were recorded in deuterated solvents and the chemical shifts are relative to SiMe₄ for ¹H, ¹³C and ²⁹Si; BF₃.Et₂O for ¹¹B, respectively. The following abbreviations are used to describe signal multiplicities: s = singlet, d = doublet, m = multiplet, brs = broad singlet. Coupling constants J are given in Hertz (Hz). HRMS spectra were obtained at the Mass Spectrometry Laboratory in the Division of Chemistry and Biological Chemistry, Nanyang Technological University. Melting points were measured with an OptiMelt automated melting point system. Fourier transform infrared (FT-IR) spectra were recorded on a Bruker Alpha FT-IR spectrometer.

**Synthesis of (TMS)₂NBBr₂ adapted from a published procedure[32].**

A hexane solution (2.5 M) of n-BuLi (8.0 mL, 20 mmol) was added dropwise into a hexane solution of hexamethyldisilazane (4.19 mL, 20 mmol) at −78 °C. The mixture was allowed to warm to room temperature and stirred for 4 h, then cooled to −78 °C, to which a hexane solution of BBr₃ (1.90 mL, 20 mmol) was added dropwise. The mixture was gradually warmed to room temperature and stirred overnight. The resulting suspension was filtered, and all volatiles were removed in vacuo to give a yellow liquid. Distillation afforded (TMS)₂NBBr₂ as a colorless liquid in 32 % yield (2.12 g, 6.65 mmol).

**Synthesis of MesBBr₂ adapted from a published procedure[33].**

A toluene solution of BBr₃ (0.95 mL, 10 mmol) was added dropwise into a toluene solution of mesitylcopper(I) (2.01 g, 10 mmol) at −78 °C. The mixture was stirred for 2 h at −78 °C before it was allowed

to warm to room temperature and stirred overnight. The resulting suspension was filtered, and all volatiles were removed *in vacuo* to give a yellow liquid. Distillation afforded MesBBr$_2$ as a colorless liquid in 72 % yield (2.08 g, 7.17 mmol).

## Synthesis of 1

A toluene solution of (TMS)$_2$NBBr$_2$ (1.0 mmol) was added into a 100 mL Schlenk flask containing 1,3,4,5-tetramethylimidazolin-2-ylidene (2.0 mmol, 0.25 g) and KC$_8$ (2.0 mmol, 0.27 g) at room temperature, following which, the reaction mixture was stirred for 8 h. The resulting bright red-purple suspension was filtered, and the filtrate was concentrated to 10 mL and kept for 3 days at room temperature to afford compound **1** as red block crystals (0.31 g) in 73 % yield. M.p.: 76 °C. $^1$H NMR (399.5 MHz, C$_6$D$_6$, 25 °C): δ 3.27 (s, 6 H, N-C$H_3$), 2.41 (s, 6 H, N-C$H_3$), 1.66 (s, 12 H, C-C$H_3$), 0.44 (s, 18 H, N(Si(C$H_3$)$_3$)$_2$). $^{11}$B{$^1$H} NMR (128 MHz, C$_6$D$_6$, 25 °C): δ 1.6 (s). $^{13}$C{$^1$H} NMR (101 MHz, C$_6$D$_6$, 25 °C): δ 121.3, 119.8 (*C* = *C*), 34.9, 34.8 (N*C*H$_3$), 10.0, 9.3 (*C*H$_3$), 4.7 (N(Si(*C*H$_3$)$_3$)). $^{29}$Si{$^1$H} NMR (79.4 MHz, C$_6$D$_6$, 25 °C): δ −1.5 (s). HRMS (ESI): m/z calcd for C$_{20}$H$_{43}$BN$_5$Si$_2$: 420.3150 [(M + H)]$^+$; found: 420.3157.

## Synthesis of 2

A toluene solution of dibromo(2,4,6-trimethylphenyl)borane (MesBBr$_2$) (1.0 mmol) was added into a 100 mL Schlenk flask containing 1,3,4,5-tetramethylimidazolin-2-ylidene (2.0 mmol, 0.25 g) and KC$_8$ (2.0 mmol, 0.27 g) at room temperature, following which, the reaction mixture was stirred for 8 h. The resulting bright red suspension was filtered, and the filtrate was concentrated to 10 mL and kept for 3 days at room temperature to afford compound **2** as red block crystals (0.16 g) in 42 % yield. M.p.: 94 °C. $^1$H NMR (399.5 MHz, C$_6$D$_6$, 25 °C): δ 7.20 (s, 2 H, Ar$H$), 2.75 (s, 6 H, N-C$H_3$), 2.72 (s, 6 H, N-C$H_3$), 2.57 (s, 6 H, Ar-C$H_3$), 2.47 (s, 3 H, Ar-C$H_3$), 1.72 (s, 6 H, C-C$H_3$), 1.55 (s, 6 H, C-C$H_3$). $^{11}$B{$^1$H} NMR (128 MHz, C$_6$D$_6$, 25 °C): δ − 6.8 (s). $^{13}$C{$^1$H} NMR (101 MHz, C$_6$D$_6$, 25 °C): δ 141.5, 130.9, 127.5 (Ar-*C*), 120.3, 118.8 (*C* = *C*), 34.9, 33.8 (N*C*H$_3$), 25.1, 21.7 (Ar-*C*H$_3$), 10.0, 9.2 (*C*H$_3$). HRMS (ESI): m/z calcd for C$_{23}$H$_{36}$BN$_4$: 379.3033 [(M + H)]$^+$; found: 379.3035.

## Synthesis of 3

A toluene solution of **1** (0.13 g, 0.3 mmol) in a Schlenk flask was degassed by a freeze−pump−thaw method. Then, CO$_2$ (1 bar) was filled. The resulting solution changed from red-purple to colorless immediately. After 30 min of stirring, all volatiles of the resulting suspension were removed under vacuum to give **3** as a colorless solid (0.08 g) in 80% yield. Colorless crystals of **3** were isolated from the saturated acetonitrile solution. M.p.: 81 °C. $^1$H NMR (399.5 MHz, CD$_3$CN, 25 °C): δ 3.52 (s, 12 H, N-C$H_3$), 2.14 (s, 12 H, C-C$H_3$). $^{11}$B{$^1$H} NMR (128 MHz, CD$_3$CN, 25 °C): δ −16.0 (br). $^{13}$C{$^1$H} NMR (101 MHz, CD$_3$CN, 25 °C): δ 126.7 (*C* = *C*), 33.2 (N-C$H_3$), 8.9 (*C*H$_3$). HRMS (ESI): m/z calcd for C$_{16}$H$_{25}$BN$_5$O$_3$: 346.2050 [(M + H)]$^+$; found: 346.2056.

## Synthesis of 4

A CD$_3$CN solution of **2** (0.04 g, 0.1 mmol) in a J-Young NMR tube was degassed by a freeze−pump−thaw method. Then, CO$_2$ (1 bar) was filled. The resulting solution changed from red to colorless immediately. $^1$H NMR (399.5 MHz, CD$_3$CN, 25 °C): δ 6.68 (s, 2 H, Ar$H$), 3.29 (s, 12 H, N-C$H_3$), 2.18 (s, 3 H, Ar-C$H_3$), 2.13 (s, 12 H, C-C$H_3$), 1.96 (s, 6 H, Ar-C$H_3$). $^{11}$B{$^1$H} NMR (128 MHz, CD$_3$CN, 25 °C): δ − 15.7 (s). $^{13}$C{$^1$H} NMR (101 MHz, CD$_3$CN, 25 °C): δ 144.4, 135.0, 130.2 (Ar-*C*), 126.4 (*C* = *C*), 34.1 (N-C$H_3$), 24.3, 20.7 (Ar-C$H_3$), 9.2 (*C*H$_3$).

## Synthesis of 4·B(OH)$_3$

A THF solution of **2** (0.15 g, 0.4 mmol) in a 100 mL Schlenk flask was degassed by a freeze−pump−thaw method. Then, CO$_2$ (1 bar) was filled. The reaction mixture was stirred for 30 min at room temperature. Boric acid B(OH)$_3$ (0.03 g, 0.5 mmol) was then added into the colorless solution. After which, the reaction mixture was stirred for 2 h. The

resulting suspension was filtered and concentrated to give compound **4·B(OH)$_3$** as colorless crystals (0.11 g) in 53% yield. M.p.: 93 °C. $^1$H NMR (399.5 MHz, CD$_3$CN, 25 °C): δ 6.71 (s, 2 H, Ar$H$), 3.25 (s, 12 H, N-C$H_3$), 2.18 (s, 3 H, Ar-C$H_3$), 2.14 (s, 12 H, C-C$H_3$), 1.92 (s, 6 H, Ar-C$H_3$). $^{11}$B{$^1$H} NMR (128 MHz, CD$_3$CN, 25 °C): δ 19.7 (s, *B*(OH)$_3$), − 15.8 (s, Ar-*B*). $^{13}$C{$^1$H} NMR (101 MHz, CD$_3$CN, 25 °C): δ 144.4, 134.8, 130.2 (Ar-*C*), 126.3 (*C* = *C*), 34.0 (N-*C*H$_3$), 24.3, 20.7 (Ar-*C*H$_3$), 9.2 (*C*H$_3$). HRMS (ESI): m/z calcd for C$_{24}$H$_{39}$B$_2$N$_4$O$_5$: 485.3107 [(M + H)]$^+$; found: 485.3121.

## Synthesis of 5

A toluene solution of **2** (0.15 g, 0.4 mmol) in a 100 mL Schlenk flask was degassed by a freeze−pump−thaw method. Then, CO$_2$ (1 bar) was filled. The reaction mixture was stirred for 2 h at room temperature. Ammonia borane (NH$_3$BH$_3$) (0.012 g, 0.4 mmol) was then added into the colorless solution. After which, the reaction mixture was stirred for 4 h. The resulting suspension was filtered and concentrated to give compound **5** as colorless crystals (0.09 g) in 51% yield. M.p.: 262 °C. $^1$H NMR (399.5 MHz, CDCl$_3$, 25 °C): δ 6.75 (s, 2 H, Ar$H$), 3.28 (s, 12 H, N-C$H_3$), 2.23 (s, 3 H, Ar-C$H_3$), 2.22 (s, 12 H, C-C$H_3$), 1.77 (s, 6 H, Ar-C$H_3$). $^{11}$B NMR (128 MHz, CDCl$_3$, 25 °C): δ − 23.3 (d, J = 84.0 Hz). $^{13}$C{$^1$H} NMR (101 MHz, CDCl$_3$, 25 °C): δ 167.7 (*C* = O), 141.8, 135.7, 129.3 (Ar-*C*), 126.5 (*C* = *C*), 32.6 (N-C$H_3$), 23.4, 21.0 (Ar-C$H_3$), 9.3 (*C*H$_3$). HRMS (ESI): m/z calcd for C$_{24}$H$_{37}$BN$_4$O$_2$: 441.3037 [(M + H)]$^+$; found: 441.3038.

## Synthesis of 6

A toluene solution of **2** (0.15 g, 0.4 mmol) in a 100 mL Schlenk flask was degassed by a freeze−pump−thaw method. Then, CO$_2$ (1 bar) was filled. The reaction mixture was stirred for 2 h at room temperature. Aniline (PhNH$_2$) (0.037 g, 0.4 mmol) was then added into the colorless solution. After which, the reaction mixture was stirred for 4 h. The resulting suspension was filtered and concentrated to give compound **6** as colorless crystals (0.12 g) in 72 % yield. M.p.: 229 °C. $^1$H NMR (399.5 MHz, CDCl$_3$, 25 °C): δ 7.37 (d, 1 H, Ar$H$, $^3$J$_{H-H}$ = 7.6 Hz), 7.13-7.09 (m, 1 H, Ar$H$), 7.06-7.02 (m, 1 H, Ar$H$), 6.72 (s, 2 H, Ar$H$), 6.72-6.62 (m, 2 H, Ar$H$), 3.21 (s, 12 H, N-C$H_3$), 2.19 (s, 3 H, Ar-C$H_3$), 2.15 (s, 12 H, C-C$H_3$), 1.72 (s, 6 H, Ar-C$H_3$). $^{11}$B NMR (128 MHz, CDCl$_3$, 25 °C): δ − 23.4 (d, J = 81.7 Hz). $^{13}$C{$^1$H} NMR (101 MHz, CDCl$_3$, 25 °C): δ 162.4 (*C* = O), 158.9, 146.5, 144.3, 141.7, 135.5, 129.3, 129.2, 128.2 (Ar-*C*), 125.9 (*C* = *C*), 118.5, 118.1, 116.6, 115.1(Ar-*C*), 32.4 (N-C$H_3$), 23.3, 20.9 (Ar-C$H_3$), 9.1 (*C*H$_3$). HRMS (ESI): m/z calcd for C$_{30}$H$_{42}$BN$_5$O$_2$: 441.3037 [(M + H)]$^+$; found: 441.3038.

## Data availability

All data generated or analyzed during this study are included in this manuscript (and its Supplementary Information). Details about materials and methods, experimental procedures, characterization data, and NMR spectra are available in the Supplementary Information. The optimized cartesian coordinates are provided in the Source Data file. The structures of **1**−**6** in the solid state were determined by single-crystal X-ray diffraction studies and the crystallographic data for these structures have been deposited at the Cambridge Crystallographic Data Centre (CCDC) under deposition numbers 2235472 (**1**), 2257718 (**2**), 2235466 (**3**), 2257719 (**4·B(OH)$_3$**), 2307337 (**5**), 2307338 (**6**). These data can be obtained free of charge from via www.ccdc.cam.ac.uk/ data_request/cif. All data are also available from corresponding authors upon request. Source data are provided with this paper.

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

## Acknowledgements

This work was supported by the Ministry of Education Singapore, AcRF Tier 1 (RG72/12) and A*STAR MTC Individual Research Grants (M21K2c0117) (C.-W.S.). C.-S.W. and M.-D.S. acknowledge the National Center for High-Performance Computing of Taiwan for generous amounts of computing time and the Ministry of Science and Technology of Taiwan for the financial support. We also thank Prof. Weng Kee Leong and Zhen Xuan Wong (NTU) for their assistance in Fourier transform infrared spectroscopy, as well as Dr. Yongxin Li (NTU) for his assistance in the X-ray crystallographic measurements and analysis.

## Author contributions

J.F. and A.-P.K. performed the synthetic experiments and spectroscopic characterizations. C.-S.W. and M.-D.S. did the theoretical calculations. C.-W.S. conceived and supervised the study, and drafted the manuscript with assistance from J.F. and A.-P.K. All authors contributed to discussion.

## Competing interests

The authors declare no competing interests.
