## [Peer Review File · Nature Communications]

Carbon Dioxide Capture and Functionalization by Bis(N-Heterocyclic Carbene)-Borylene ComplexesReviewers' Comments:

Reviewer #1:

Remarks to the Author:

the manuscript submitted by Su, So and co-workers reported the synthesis of Bis(N-Heterocyclic Carbene)-Borylene Complexes and their application in CO₂ capture and functionalization. Almost all borylene structures were isolated and well-defined by X-ray chromatograph. Especially, the key intermediate 3-bis(NHC)-isocyanatoborylene-carbon dioxide adduct, which is challenging, and has been isolated and characterized by X-ray chromatograph. Additionally, the electronic structure of 2 and 3 has been elucidated based on a DFT calculation, such as frontier orbital analysis, and bond order analysis. Meanwhile, the process of CO₂ activation was also studied by computational study. Overall, this is a solid work, and this reviewer recommends its publication in this journal after the following minor revisions:

1. Regarding the choice of calculation level, especially for the study of the reaction mechanism of carbon dioxide activation, the selected B3LYP-D3(BJ)/def2SVP theory of level is too low. It is not very suitable to use for the functional. M062X with a triple-zeta basis set, such as M06-2x/def2-TZVP, should be selected as the optimal choice for calculating the electronic energy.
2. The isovalue of the orbital surface should be noted in the footnote or SI.
3. the DFT calculations should consider the solvent model or the author should note the gas phase or the solvent model.
- 4 . The layout of the pictures in the article is a bit messy, and there are some small problems that should be avoided to increase the beauty of the article. For example, there are no spaces on either side of the equal sign in Figure 3 and so on.

Reviewer #2:

Remarks to the Author:

Professor So and coworkers have presented the synthesis of two new bis(carbene)borylenes and their reactivity with CO₂. The first examples of boron bound CO₂ featuring an η¹ κC coordination mode are crystallographically characterized, and are of significant fundamental interest. The stoichiometric functionalization of borylene-bound CO₂ with NH₃BH₃ and aniline is also presented. The work described herein is well supported by spectroscopic and crystallographic data. The crystallographic data is of good quality and is well-treated. The mechanism for the formation of the isocyanato-borylene CO₂ adduct has been studied computationally and supports its isolation.

However, the mode of stoichiometric CO₂ hydrogenation presented in this manuscript is well known with ammonia-borane in the presence of a strong base. Given that low valent boron, in the form of a diazadiborane, has recently been reported to catalyze the N-formylation of amines with CO₂, I don't believe that the impact of this work rises to the bar of Nature Communications. However, I would enthusiastically support its publication another journal such as *Angewandte Chemie* or *Chemical Science*.

A few additional points to consider:

In the mechanism, have the authors also considered elimination of hexamethyldisiloxane from Int2 to form and isocyanato-borylene intermediate, which then could in turn activate additional CO₂?

The mimicry of CO dehydrogenase by compound 4 is somewhat overstated. Hydrogen bonding in carbonates is well known.

Minor corrections:

Line 12-13 "...enabling the latter to be functionalized by disilylamido to form..." is somewhat

confusingly worded. perhaps something like "..functionalized by the boron-bound disilylamido moiety to form.." would be clearer

Line 80. Include esd on the dihedral angle

Line 223. The word 'activates' is used to describe the scission of an O-H bond in the deprotonation of a carboxylic acid by a borylene. I would argue that this reaction likely deactivates these atoms toward further reactivity.

Reviewer #3:

Remarks to the Author:

The manuscript from Su, So and coworkers presents two highly nucleophilic borylenes with two NHCs. They then treat them with CO₂, whereby the borylene attacks at the carbon atom of the CO₂, an unprecedented reactivity pattern. In further reactivity, they hydrogenate and functionalize the CO₂, both unknown reactions with borylenes. The synthesis and characterization appears to have been done competently.

The work thus presents the first examples of a long-awaited class of compounds (bis(NHC) borylenes), expected to be very electron-rich species. They also demonstrate likely the first true example of a monovalent boron atom acting as a strong nucleophile. As such, the work is a significant achievement, and I believe worthy of publication in Nature Communications. I have a few small points to raise before acceptance, however, as listed below.

- Introduction: some mention of conventional FLPs might be in order here, since the nucleophilic part of a conventional FLP attacks the C atom of CO₂. That the boron is usually the electrophilic part of an FLP makes this work a nice example of "umpolung".

- The abstract doesn't seem to provide much insight and doesn't seem to fit with the journal's normal style. It is more full of compound formulae and compound numbers, rather than an insightful description of what is happening.

- The conclusions are similar to the abstract - lots of formulae and no insight. A summary of the advances (first bis(NHC) borylene, first single-site CO₂ activation with boron, etc.) would be much more useful here than merely a summary of the reactions performed.

Dear Reviewers,

Thank you for reviewing the manuscript. It has been revised according to the Author Checklist as well as the following Reviewers' comments.

Reviewer 1

1. **Comments:** Regarding the choice of calculation level, especially for the study of the reaction mechanism of carbon dioxide activation, the selected B3LYP-D3(BJ)/def2SVP theory of level is too low. It is not very suitable to use for the functional. M062X with a triple-zeta basis set, such as M06-2x/def2-TZVP, should be selected as the optimal choice for calculating the electronic energy.

Reply: We have recalculated the electronic energies using the suggested level of theory and basis set.

2. **Comments:** The isovalue of the orbital surface should be noted in the footnote or SI.

Reply: The isovalue of 0.06 has been noted in both the manuscript and SI. Thank you for the reminder.

3. **Comments:** the DFT calculations should consider the solvent model or the author should note the gas phase or the solvent model.

Reply: The solvent model has been considered and noted.

4. **Comments:** The layout of the pictures in the article is a bit messy, and there are some small problems that should be avoided to increase the beauty of the article. For example, there are no spaces on either side of the equal sign in Figure 3 and so on.

Reply: We have edited the layout of some of the pictures and the mentioned problem has been rectified. Thank you for the suggestions.

Reviewer 2

1. **Comments:** In the mechanism, have the authors also considered elimination of hexamethyldisiloxane from Int2 to form and isocyanato-borylene intermediate, which then could in turn activate additional CO₂?

Reply: We have indeed considered such a mechanism, which we found to be infeasible. Below is DFT-calculated free energy profile for proposed mechanism via elimination of hexamethyldisiloxane.

2. **Comments:** The mimicry of CO dehydrogenase by compound **4** is somewhat overstated. Hydrogen bonding in carbonates is well known.

Reply: We disagree with this statement. First, compound **4** is a carboxylate and not a carbonate. Next, captured CO₂ tends to require stabilization using a strong Lewis acid. For example, in the activation of CO₂ by an FLP of PtBu₃ and B(C₆F₅)₃, the phosphine acts as a Lewis base to capture CO₂ while the Lewis acidic borane stabilizes the captured CO₂ via O-coordination. As such, our case is novel, because only hydrogen bonding was required to stabilize the captured CO₂. [Ref: Mömming, C., Otten, E., Kehr, G., Fröhlich, R., Grimme, S., Stephan, D. and Erker, G. Reversible Metal-Free Carbon Dioxide Binding by Frustrated Lewis Pairs. *Angew. Chem. Int. Ed.* **48**, 6643-6646, (2009).]

3. **Comments:** Line 12-13 "...enabling the latter to be functionalized by disilylamido to form..." is somewhat confusingly worded. perhaps something like "...functionalized by the boron-bound disilylamido moiety to form..." would be clearer

Reply: We have edited the abstract to be more insightful and clearer. Thank you for your suggestion.

4. **Comments:** Line 80. Include esd on the dihedral angle

Reply: The esd on the dihedral angle has been included.

5. **Comments:** Line 223. The word 'activates' is used to describe the scission of an O-H bond in the deprotonation of a carboxylic acid by a borylene. I would argue that this reaction likely deactivates these atoms toward further reactivity.

Reply: We disagree with this statement. Compound **5**, which is formed after the scission of the O-H bond, can undergo further reaction with an alkyne. The carboxylate group is introduced to the alkyne and the borylene is regenerated. Such reactivity will be reported in due time.

Reviewer 3

1. **Comments:** Introduction: some mention of conventional FLPs might be in order here, since the nucleophilic part of a conventional FLP attacks the C atom of CO₂. That the boron is usually the electrophilic part of an FLP makes this work a nice example of "umpolung".

Reply: We have included a short description of FLPs in the introduction to make a comparison. Thank you for the suggestion.

2. **Comments:** The abstract doesn't seem to provide much insight and doesn't seem to fit with the journal's normal style. It is more full of compound formulae and compound numbers, rather than an insightful description of what is happening.

Reply: We have edited the abstract to provide more insight and fit with the journal's normal style. Thank you for the reminder.

3. **Comments:** The conclusions are similar to the abstract - lots of formulae and no insight. A summary of the advances (first bis(NHC) borylene, first single-site CO₂ activation with boron, etc.) would be much more useful here than merely a summary of the reactions performed.

Reply: We have edited the conclusion to provide a summary of the advances. Thank you for the suggestion.

Thank you for your kind consideration.

Yours faithfully,

Cheuk-Wai So

Reviewers' Comments:

Reviewer #1:

Remarks to the Author:

After careful review and consideration of the revisions made to this manuscript, I am pleased to recommend its acceptance for publication.

Reviewer #2:

Remarks to the Author:

Professor So and coworkers have presented the first isolation of η^1 -CO₂- κ^C borylene complexes, and subsequent functionalization reactions with ammonia-borane and aniline.

Generally, the compounds are well characterized and the crystallographic data is well treated, the CO₂ complexes presented are of significant fundamental interest, and the responses to reviewer comments seem for the most part reasonable. However, I have a concern. At several points in the text (line 20, 72, and 228), regeneration of **2** by further reaction of coordinated CO₂ in **4** is implied, which in turn is used to claim the viability of **2** in catalytic transformations with CO₂. It appears that this reasoning is based on speculation, and the authors have not presented evidence for the reformation of **2** in these reactions. In the absence of data or computational support, the authors should probably avoid speculating on this point, or use much softer language in doing so. The regeneration of **2** in these reactions is even mentioned in the abstract, despite the lack of evidence for it. Certainly, the strongest proof would be in the form of a report of compound **2**'s catalytic activity.

Also, I find the writing to be awkward in a few places:

87 "The planes of the planar boron center and planar nitrogen center form a dihedral angle of 58(1) ° is awkwardly stated. Maybe: "The Si-N-B-C dihedral angle is 58(1)°"

115 it seems a bit strange to say that a pi-acidic substituent is 'essential' to the stability of bis(NHC)borylenes and then present two examples to the contrary

131. 'was used to react with' vs. 'was reacted with'

131-45. I feel this paragraph could benefit from some reorganization. Perhaps it would be clearer to discuss analytical and metrical parameters first and then discuss the mechanism? The abrupt transition in line 140 from mechanism to characterization without segue detracts from readability.

162 awkward writing: "signifying that a borylene alone can provide direct access to incorporate boron at the carbon center of CO₂ which is infeasible in the chemistry of electrophilic borane with CO₂"

195-199 "Based on this, *an* effective single-site catalyst motif, where ligands with pendant proton donors are used to coordinate with a transition metal for the activation of CO₂ through the stabilization of transition metal-CO₂ adduct by hydrogen bonding, has been developed in biomimetic artificial CO₂ reduction catalysis.

247 is this supposed to be "Conclusion"?

Dear Reviewers,

Thank you for reviewing the manuscript. It has been revised according to the Author Checklist as well as the following Reviewers' comments. Please kindly refer to the manuscript file with tracking for your reference.

DFT calculations were performed to demonstrate that regeneration of compound **2** in the functionalization of the captured CO₂ is feasible. The DFT calculations are shown in Figure 5b and Supplementary Fig. 39. The molecular structures of **4**·B(OH)₃, **5** and **6** are given in Figure 6. The Figure captions and manuscript are revised accordingly.

Reviewer 2

1. **Comments:** At several points in the text (line 20, 72, and 228), regeneration of **2** by further reaction of coordinated CO₂ in **4** is implied, which in turn is used to claim the viability of **2** in catalytic transformations with CO₂. It appears that this reasoning is based on speculation, and the authors have not presented evidence for the reformation of **2** in these reactions. In the absence of data or computational support, the authors should probably avoid speculating on this point, or use much softer language in doing so. The regeneration of **2** in these reactions is even mentioned in the abstract, despite the lack of evidence for it. Certainly, the strongest proof would be in the form of a report of compound **2**'s catalytic activity.

Reply: We have removed the mention of regeneration of **2** in lines 20 and 70 but kept the one in line 227 in the revised manuscript. The mention in line 227 refers to a proposed mechanism, which is now supported by DFT calculations (Figure 5b, Supplementary Fig. 39). We have edited the sentences (lines 227 – 232) relating to the mechanism to reflect this. Thank you for the suggestions.

2. **Comments:** 87 “The planes of the planar boron center and planar nitrogen center form a dihedral angle of 58(1)° is awkwardly stated. Maybe: “The Si-N-B-C dihedral angle is 58(1)°”

Reply: “The Si-N-B-C dihedral angle is 58(1)°” is stated in line 85.

3. **Comments:** 115 it seems a bit strange to say that a pi-acidic substituent is ‘essential’ to the stability of bis(NHC)borylenes and then present two examples to the contrary

Reply: The two examples presented both feature π -acidic substituents, which are the Cym ligand and isocyanide ligand. The examples serve to demonstrate that in previously reported bis(NHC)borylenes with two weak π -accepting NHC ligands, the third ligand has to be π -electronic withdrawing. We have edited the phrasing in that paragraph (line 113 – 115) to better explain this. “essential” is removed in the revised manuscript.

4. **Comments:** 131. ‘was used to react with’ vs. ‘was reacted with’

Reply: “was reacted with” is stated in line 130.

5. **Comments:** 131-45. I feel this paragraph could benefit from some reorganization. Perhaps it would be clearer to discuss analytical and metrical parameters first and then discuss the mechanism? The abrupt transition in line 140 from mechanism to characterization without segue detracts from readability.

Reply: We have reorganized the paragraph. Discussion of analytical data is written (Line 132) before the mechanism. There is no change in the content.

6. **Comments:** 162 awkward writing: “signifying that a borylene alone can provide direct access to incorporate boron at the carbon center of CO₂ which is infeasible in the chemistry of electrophilic borane with CO₂”

Reply: We have edited the sentence to flow better (lines 160 – 162):

“The formation of **3** demonstrates that the boron center in a borylene can directly attack the carbon center of CO₂, which is a result that has remained unattainable using electrophilic borane.”

7. **Comments:** 195-199 “Based on this, *an* effective single-site catalyst motif, where ligands with pendant proton donors are used to coordinate with a transition metal for the activation of CO₂ through the stabilization of transition metal-CO₂ adduct by hydrogen bonding, has been developed in biomimetic artificial CO₂ reduction catalysis.

Reply: “an” is added in line 198.

8. **Comments:** 247 is this supposed to be “Conclusion”?

Reply: Line 247 (251 in the revised manuscript) is “Discussion”, which is the heading advised by the editor in the Author’s Checklist.

Thank you for your kind consideration.